# Bayesian Learning with Information Gain Provably Bounds Risk for a Robust Adversarial Defense

## Abstract

In this paper, we present a novel method to learn a Bayesian neural network robust against adversarial attacks. Previous algorithms have shown an adversarially trained Bayesian Neural Network (BNN) provides improved robustness against attacks. However, the learning approach for approximating the multi-modal Bayesian posterior leads to mode collapse with consequential sub-par robustness and under performance of an adversarially trained BNN. Instead, we propose approximating the multi-modal posterior of a BNN to prevent mode collapse and encourage diversity over learned posterior distributions of models to develop a novel *adversarial training method for BNNs*. Importantly, we conceptualize and formulate information gain (IG) in the adversarial Bayesian learning context and prove, training a BNN with IG bounds the difference between the conventional empirical risk with the risk obtained from adversarial training—our intuition is that information gain from data and adversarial examples should be the same for a robust BNN. Extensive experimental results demonstrate our proposed algorithm to achieve state-of-the-art performance under strong adversarial attacks.

## 1 Introduction

Deep neural networks (DNNs) have demonstrated *impressive* performance on multiple tasks, such as image recognition (He et al., 2016) or natural language processing (Vaswani et al., 2017). However, despite the impressive performance, DNNs are poor at quantifying the predictive uncertainty and tend to produce overconfident predictions. Consequently, it has been shown to be vulnerable to adversarial examples (AEs) (Goodfellow et al., 2015; Madry et al., 2018; Carlini & Wagner, 2017) where carefully crafted perturbations are added to the inputs to drive the performance of DNNs. These perturbations are *imperceptible* to human eyes (Goodfellow et al., 2015) (in the context of image classification) but can drastically degrade the performance of the DNNs. There are various methods to find such perturbations (Madry et al., 2018; Goodfellow et al., 2015; Carlini & Wagner, 2017; Papernot et al., 2016a). Importantly, these threats are also shown to be effective in the physical world (Kurakin et al., 2018; Eykholt et al., 2018) and are shown to be effective in transferring across models to make *black-box* attacks (Papernot et al., 2016a; 2017). Therefore, adversarial perturbations present a realistic threat for DNN applications and motivate the need to develop robust DNNs.

Despite the huge amount of effort to overcome the thread from AEs since its first introduction in 2014 (Szegedy et al., 2014), training a robust DNN against AEs is challenging. Athalye et al. (2018) have shown that one of the most robust methods to defend against this threat is Adversarial Training (Madry et al., 2018). In this method, the network is trained with adversarial examples in order to create robustness against the AEs at inference time. Nevertheless, as mentioned in Ye & Zhu (2018), this method relies on the "point estimate" approach of the deep neural network ; hence it lacks the capability to deal with the uncertainty of the adversary at inference time *i.e.* adversaries beyond the pre-defined norm. In addition, using a point estimate only defines a single decision boundary that could be manipulated easily with an adversarial input. Alternatively, one can use multiple decision boundaries, or more precisely, integrate out the effects of parameters in the model. That is the premise of Bayesian learning methods Welling & Teh (2011) that define a distribution over the parameters leading to Bayesian Deep Neural Networks (BNNs). Thus, the output of *predictive* distribution is obtained by integrating out the parameters w.r.t. their distribution.

Motivated by the intuition that removing the effects of the parameter choice can lead to more robust models, (Liu et al., 2019) proposed adversarial training of BNNs and demonstrated impressive results. However, training BNNs pose a significant challenge: the exact solution of posterior distribution (*i.e.* the parameter distribution after observing the data) is often *intractable*. Efforts devoted to developing a suitable inference approach to approximate the posterior involve either using Markov Chain Monte Carlo (MCMC; asymptotically accurate but slow; see *e.g.* (Welling & Teh, 2011)) or variational inference (efficient but inaccurate; see *e.g.* (Blei et al., 2017)). For instance, Liu et al. (2019) uses variational methods (Blei et al., 2017; Blundell et al., 2015) to approximate the posterior with a unimodal Gaussian distribution. The issue with such a method for deep learning is that the parameters sampled are in the proximity of the mode and, consequently, does not capture the multi-modal aspect of the distribution. This leads to attaining only minor variations in parameters and sub-par robustness.

In this study, inspired by BNNs, we consider a novel approach for robust learning. We hypothesize *(1)* a model that relies on the parameter distribution rather than a single point estimate which *(2)* predicts the same *predictive distribution* for both the given dataset and its adversarials, is more robust. Inspired by (Liu & Wang, 2016), to achieve (1) to remedy the lack of diversity in parameters we exploit Stein Variational Gradient Descent (SVGD). SVGD marries MCMC and variational inference to attain the benefits of both. It leads to sampling a set of parameter "particles" that are encouraged to be *diverse* to fit the modes of the true posterior distribution. Further, by utilizing this approach for achieving (2), we compute the *Information Gain* (IG) for each input instance. In other words, we measure value of knowing the label for an input given the posterior distribution for either the given benign training set versus the adversarial counterparts. This simply translates to matching the predictive distribution of the benign to the adversarial instances.

Given our intuitions, we designed an objective that satisfies our hypothesis. Further, show that our conceptualization and formulation of information gain (IG) in the adversarial Bayesian learning context bounds the difference between the conventional empirical risk with the risk obtained from adversarial training. This is the *first* such bound that establishes the relation between Bayesian learning with information gain to the generalization of empirical risk minimization (ERM). We summarize our results and contributions below:

- We propose a novel method to learn a BNN robust against adversarial attacks that utilizes: i) the information gain of the instances from the dataset; and ii) SVGD to generate parameter particles that are trained in parallel to be as diverse as possible while maintaining the same measure of information content for benign and adversarial training instances. Our learning approach enables the model to both reduce the effect of single parameter choice and learn the invariant patterns that are common between the training dataset and its corresponding adversarial samples.

- We show that learning a BNN with an Information Gain formulation in the Bayesian context yields an upper bound on the difference between the empirical risk versus adversarial risk. As such, minimizing the learning objective we propose, provably approaches the same bounds as empirical risk minimization. This is the first time such a bound is devised and is significant as it provides a theoretically justified approach to reducing the uncertainty due to adversarial examples.

- In the adversarial training approach we propose for BNNs, in contrast to using the predictive distribution (*i.e.* as opposed to using the whole posterior distribution) to obtain the adversarial instances using PGD, we use samples from the posterior. This method, not only allows for a more efficient implementation, but also encourages the parameter particles to remain diverse and generate dissimilar adversarial patterns. The result is a more effective approach to improve robustness during training.

- Comprehensive evaluations on a set of neural architectures and datasets demonstrates our approach to result in up to 20% improvement compared to the state-of-the-art in the robustness.

## 2 BACKGROUND & RELATED WORK

**Primer on Bayesian Learning**. Given a dataset $\mathcal{D} = \{\mathbf{x}_i, y_i\}_{i=1}^{N}$, a Bayesian Neural Network (BNN) aims to learn the *posterior* distribution: $p(\boldsymbol{\theta} \mid \mathcal{D}) = \frac{p(\mathcal{D}|\boldsymbol{\theta})p(\boldsymbol{\theta})}{p(\mathcal{D})}$ given the prior distribution $p(\boldsymbol{\theta})$. However, the exact solution for the posterior is often *intractable* due to the high dimensional integral of the denominator even for moderately sized networks in the context of deep learning (Blei

et al., 2017). To overcome this, there are generally two approaches. The first one is to use Markov Chain Monte Carlo (MCMC) methods, which shows to be asymptotically accurate but slow (Welling & Teh, 2011). Alternatively, one can use variational inference methods to approach the true posterior distribution (Blundell et al., 2015). However, in the case of BNN, the true Bayesian posterior is usually a complex multimodal distribution (Izmailov et al., 2021) as illustrated in Figure 1. Variational inference, which relies on a parametric function, is too restrictive to resemble the true posterior and suffers from mode collapse (Izmailov et al., 2021; Jospin et al., 2020).

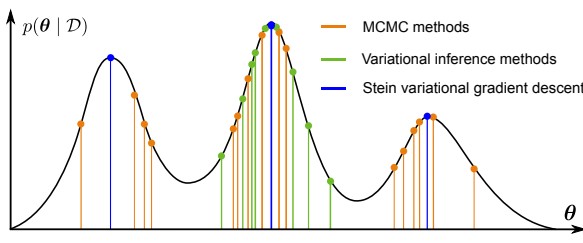

Figure 1: Different techniques to sample the posterior.

On the other hand, Wang & Liu (2019); Liu & Wang (2016) proposed a provable general purpose variational inference algorithm named Stein Variational Gradient Descent (SVGD) that transports a set of parameter particles, encouraged to be diverse, to fit the true posterior distribution; this approach can be beneficial for achieving higher performance and approximating the true posterior distribution. The visualization for different techniques to sample the posterior is displayed in Figure 1.

**Adversarial Attacks**. Attackers can add carefully crafted noise (perturbations) to the input image to fool the classifier at the inference stage. In general, the goal of the attacker—described in Equation (1)—is to degrade the performance of a neural network by crafting $\boldsymbol{\delta}$, such that:

$$\max_{\|\boldsymbol{\delta}\|_p < \varepsilon_{\max}} \ell(f(\mathbf{x} + \boldsymbol{\delta}; \boldsymbol{\theta}), y) \tag{1}$$

where, $p$ is the norm, $\varepsilon_{\max}$ is the maximum attack budget (perturbation), $\ell$ is the loss function (typically cross-entropy), $f$ is the network, $\mathbf{x}$ is the input, $\boldsymbol{\theta}$ is the network parameter, and $y$ is the ground-truth label.

For a PGD (Madry et al., 2018) attack, an attacker starts from $\mathbf{x}^0 = \mathbf{x}_o$ and conducts projected gradient descent iteratively to update the adversarial example following (2):

$$\mathbf{x}^{t+1} = \Pi_{\varepsilon_{\max}} \left\{ \mathbf{x}^t + \alpha \cdot \text{sign}\left( \nabla_{\mathbf{x}} \ell\left( f\left(\mathbf{x}^t; \boldsymbol{\theta}\right), y_o\right) \right) \right\} \tag{2}$$

where $\Pi_{\varepsilon_{\max}}$ is the projection to the set $\{\mathbf{x} \mid \|\mathbf{x} - \mathbf{x}_o\|_\infty \leq \varepsilon_{\max}\}$

Among all the attack methods, we decided to apply PGD in our experiments because: i) PGD (Madry et al., 2018) is regarded as the strongest attack in terms of the $\ell_\infty$ norm and ii) it gives us direct control over the distortion by changing $\varepsilon_{max}$. However, Liu et al. (2019) showed that one cannot directly apply a PGD attack in a BNN setting. Instead, the authors suggest using a stochastic approach and proposed an updated PGD method, described in (3), for which they sample stochastic parameters $\boldsymbol{\theta}^t$ in each update step $t$.

$$\mathbf{x}^{t+1} = \Pi_{\varepsilon_{\max}} \left\{ \mathbf{x}^t + \alpha \cdot \text{sign}\left( \nabla_{\mathbf{x}} \ell\left( f\left(\mathbf{x}^t; \boldsymbol{\theta}^t\right), y_o\right) \right) \right\} \tag{3}$$

**Adversarial Defenses**. Significant research efforts describe methods to mitigate this threat, such as distillation (Papernot et al., 2016b), input denoising (Song et al., 2017) or feature denoising (Xie et al., 2019), curious readers can find more from (Kurakin et al., 2018). Among these methods, adversarial training (Madry et al., 2018) is shown to be one of the most effective and popular methods to defend against adversarial attacks. The goal of adversarial training is to incorporate the adversarial search within the training process and, thus, realize robustness against adversarial examples at test time. This is achieved by solving the following optimization problem:

$$\boldsymbol{\theta}^* = \arg\min_{\boldsymbol{\theta}} \mathbb{E}_{(\mathbf{x},y)\sim D} \left\{ \max_{\|\boldsymbol{\delta}\|_p < \varepsilon_{\max}} \mathbb{E}_{\boldsymbol{\theta}}[\ell(f(\mathbf{x} + \boldsymbol{\delta}; \boldsymbol{\theta}), y)] \right\} \tag{4}$$

where $\mathcal{D}$ is the training data. An approximate solution can be realized by generating the PGD adversarial examples from Equation (2) and then minimizing the classification loss based on the generated adversarial examples.

**Prior Art on Bayesian Defenses.** Using Bayesian Neural Networks to detect adversarial attacks was proposed in (Feinman et al., 2017; Smith & Gal, 2018). Ye & Zhu (2018) and Liu et al. (2019) tried to combine Bayesian learning with adversarial training. Particularly, Ye & Zhu (2018) present a method to jointly sample from the model's parameter posterior and the distribution of adversarial samples given the current parameter posterior. Recently, Liu et al. (2019) further developed the direction in Random Self-Ensemble (Liu et al., 2018) to build an adversarially-trained Bayesian neural network method named Adv-BNN that can scales up to complex data by adding noise to each weight instead of input or hidden features as in RSE (Liu et al., 2018). Adv-BNN also incorporates adversarial training to learn a *variational posterior distribution* to further improve model robustness against adversarial examples. However, using the variational inference method is likely to lead to mode collapse and limited the performance of the BNN (Izmailov et al., 2021) as we discussed earlier and demonstrate in our experiments in Section 4. Hence, in this work, we proposed exploring SVGD (Liu & Wang, 2016) as a Bayesian inference method to achieve a better approximation for the multi-modal posterior of a BNN. Using this approach, it is also easy to convert a traditional neural network to a Bayesian counterpart without much effort to modify the traditional neural network architecture. Further, by employing the *repulsive force* for encouraging exploration in the parameter space, we conceptualize the information gain in Bayesian learning.

## 3 METHOD

### 3.1 BAYESIAN FORMULATION FOR ADVERSARIAL LEARNING

In contrast to a point estimate learned in traditional deep learning models, in Bayesian learning, the posterior of the parameters is obtained using the Bayes rule. The posterior distribution given the dataset is defined as: $p(\boldsymbol{\theta} \mid \mathcal{D}) = \prod_{(\mathbf{x},y)\sim\mathcal{D}} p(y \mid \mathbf{x}, \boldsymbol{\theta})p(\boldsymbol{\theta})/Z'$ where $Z'$ is the normalizer. Similarly in the adversarial setting, given the adversarial dataset $\mathcal{D}_{\text{adv}}$, we have:

$$p(\boldsymbol{\theta} \mid \mathcal{D}_{\text{adv}}) = \frac{\prod_{(\mathbf{x},y)\sim\mathcal{D}} p(\mathbf{x}_{\text{adv}} \mid \mathbf{x}, y, \varepsilon_{\max}, \boldsymbol{\theta})p(y \mid \mathbf{x}, \boldsymbol{\theta})p(\boldsymbol{\theta})}{Z} \,, \tag{5}$$

where $Z$ is the normalizer, $p(\mathbf{x}_{\text{adv}} \mid \mathbf{x}, y, \boldsymbol{\theta})$ is the distribution of the Bayesian adversarial examples (we describe Bayesian adversarial examples in Section 3.2, later) and $\varepsilon_{\max}$ is the hyper-parameter for producing the adversarial. The posterior in general is intractable and we need to resort to approximations. In particular, we propose utilizing Stein variational gradient descent (SVGD) (Liu & Wang, 2016) which provides an approach to learn multiple *particles* for parameters in parallel to approximate the true posterior. SVGD uses a repulsive loss to encourage the diversity of parameter particles to prevents mode collapse. This diversity enables learning multiple models to represent various patterns in the data.

During test time, given the data point $\mathbf{x}$, we can approximate the robust Bayesian prediction with respect to the adversarial posterior using the Monte Carlo samples from $p(\boldsymbol{\theta} \mid \mathcal{D}_{\text{adv}})$

$$p(y \mid \mathbf{x}, \mathcal{D}_{\text{adv}}) = \int p(y \mid \mathbf{x}, \boldsymbol{\theta}) \, p(\boldsymbol{\theta} \mid \mathcal{D}_{\text{adv}})d\boldsymbol{\theta} \approx \frac{1}{n} \sum_{i=1}^{n} p(y \mid \mathbf{x}, \boldsymbol{\theta}_i), \quad \boldsymbol{\theta}_i \sim p(\boldsymbol{\theta} \mid \mathcal{D}_{\text{adv}}), \tag{6}$$

where $\boldsymbol{\theta}_i$ is an individual parameter particle. Notably, in practice, $p(y \mid \mathbf{x}, \boldsymbol{\theta}) = \text{softmax}(f(\mathbf{x}; \boldsymbol{\theta}))$ where $f$ is a deep neural network.

### 3.2 GENERATING BAYESIAN ADVERSARIAL EXAMPLES

We also integrate a Bayesian formulation of a PGD attack. Instead of sampling different stochastic parameters $\boldsymbol{\theta}^t$ in each PGD step—as in (Liu et al., 2019) and shown in Equation (3)—which potentially leads to unrepresentative gradient directions, we sample a random $\boldsymbol{\theta}$ from our set of parameter particles $\boldsymbol{\Theta} := \{\boldsymbol{\theta}\}_{i=1}^{n}$ for a PGD attack on each (benign) data example $\mathbf{x}$ to generate the corresponding adversarial example $\mathbf{x}_{\text{adv}}$ using:

$$\mathbf{x}^{t+1} = \Pi_{\varepsilon_{\max}} \left\{ \mathbf{x}^t + \alpha \cdot \text{sign} \left( \nabla_{\mathbf{x}} \ell \left( f\left(\mathbf{x}^t; \boldsymbol{\theta}\right), y_o\right) \right) \right\} \tag{7}$$

This equation also provides an efficient way to sample an adversarial example from the distribution $p(\mathbf{x}_{adv} \mid \mathbf{x}, y, \varepsilon_{max}, \boldsymbol{\theta})$ introduced above. Here, $\ell$ is the cross entropy.

This formulation, not only better approaches PGD in a Bayesian setting, but also leads to uncovering adversarial examples that exploit vulnerabilities of different parameter particles with different parameter choices. We expect such an approach to lead to a robust model. However, this is possible when different parameter particles are diverse samples from multiple modes of the posterior facilitated by the Stein variational inference method. On the other hand, with conventional variational inference using simple unimodal distributions, parameters are sampled within a vicinity of one of the modes that do *not* adequately capture the collective vulnerability of the model.

Using the adversarial attack formulated, we can create an adversarial dataset $\mathcal{D}_{adv}$ by perturbing the observed inputs. It is then a common practice to train the neural network using the adversarial dataset in a conventional empirical risk minimization. However, it is unknown—other than through empirical studies—how such a neural network compares to training with the original dataset. In the following, we investigate this question further to develop a new approach for a robust model.

### 3.3 CONCEPTUALIZE INFORMATION GAIN FOR BAYESIAN LEARNING

The adversarial instances are generally known to exploit the particular patterns learned by the parameters. When integrating out the parameters as in the Bayesian setting, especially under the diverse parameter particles in our approach, we implicitly remove the vulnerabilities that could arise from a single choice of a parameter. In addition, using the Bayesian setting we employ, we can formulate a notion of information gain that captures the impact of adding a new instance to a dataset on the distribution of the parameters. We show that the information gain can be defined as:

$$\mathrm{IG}(\mathbf{x}, y) = \frac{1}{p(\mathcal{D})} \left( \mathbb{E}_{\boldsymbol{\theta}}[\mathbb{H}[y|\mathbf{x}, \mathcal{D}]] - \mathbb{H}[\mathbb{E}_{\boldsymbol{\theta}}[y|\mathbf{x}, \mathcal{D}]] \right). \tag{8}$$

We provide a proof of the definition in the Appendix A. This formulation quantifies an instance's informativeness for a model given the training set.

Intuitively, the information gained from an instance is proportionate to the reduction in the expected entropy by the predictive distribution. Our conjecture is that a robust neural network quantifies the information gain from an observation the same as its adversarial counterpart. In other words, a robust model ignores the perturbation and only considers the informative content of the input. We will employ these concepts in the following learning formulation.

### 3.4 FORMULATE LEARNING A ROBUST NETWORK USING INFORMATION GAIN

We formulate the objective of our training to:

1. Learn the posterior from the *adversarial* dataset. Since we use SGVD, this corresponds to learning multiple parameter particles. This amounts to minimizing the loss subject to the repulsive constraint, *i.e.* $\mathbb{E}_{(\mathbf{x}_{adv}, y) \sim \mathcal{D}_{adv}} \left[ \mathbb{E}_{\boldsymbol{\theta} \sim (\boldsymbol{\theta} | \mathcal{D}_{adv})} [\ell(f(\mathbf{x}_{adv}; \boldsymbol{\theta}), y)] \right]$. Since the adversarial dataset is generated while training the model, it depends on the particle chosen and its parameters. To account for the vulnerability of individual parameter particle, we consider adding $\ell(f(\mathbf{x}_{adv}; \boldsymbol{\theta}_l), y)$ where the parameter particle is chosen uniformly at random, *i.e.* $\boldsymbol{\theta}_l \sim \boldsymbol{\Theta}$, and $\mathbf{x}_{adv}$ is produced using that parameter particle. Since with SGVD, we ensure the samples are diverse, we have parameter particles that explore different patterns in the input. As such, the posterior obtained from the adversarials are less likely to change with perturbations in the input and hence is more robust.

2. Achieve comparable information gain from the given dataset and that from the adversarials. Thus, ensuring: i) the information gained from data and adversarial examples is encouraged to be the same, *i.e.* $\mathbb{E}_{(\mathbf{x}, y) \sim \mathcal{D}}[\mathrm{IG}(\mathbf{x})] = \mathbb{E}_{(\mathbf{x}_{adv}, y) \sim \mathcal{D}_{adv}}[\mathrm{IG}(\mathbf{x}_{adv})]$; ii) the model to be not biased towards learning from the adversarial instances; and iii) the receptive fields are active for similar and prominent features.

Combining the above concepts using the Lagrangian method, we have the following objective:

$$L(\boldsymbol{\theta}_l) = \frac{1}{n} \sum_{k=1}^{n} \ell(f(\mathbf{x}_{adv}; \boldsymbol{\theta}_k), y) + \ell(f(\mathbf{x}_{adv}; \boldsymbol{\theta}_l), y) + \lambda[\mathrm{IG}(\mathbf{x}) - \mathrm{IG}(\mathbf{x}_{adv})], \quad \boldsymbol{\theta}_l \sim \boldsymbol{\Theta} \tag{9}$$

We summarize our proposed robust Bayesian learning approach in Algorithm 1. Here, following Liu & Wang (2016), we use the RBF kernel $k(\boldsymbol{\theta}, \boldsymbol{\theta}') = \exp\left(-\frac{\|\boldsymbol{\theta}-\boldsymbol{\theta}'\|^2}{2h^2}\right)$ and take the bandwidth $h$ to be the median of the pairwise distances of the set of parameter particles at each iteration.

---

**Algorithm 1** Bayesian adversarial inference via SVGD

---

1: **Input:** A set of initial parameter particles $\{\boldsymbol{\theta}_i^0\}_{i=1}^n$, observed data $\mathcal{D}$.
2: **Output:** A set of parameter particles $\boldsymbol{\Theta} := \{\boldsymbol{\theta}_i\}_{i=1}^n$ that approximates the true posterior distribution $p(\boldsymbol{\theta}|\mathcal{D}_{\text{adv}})$
3: **for** $(\mathbf{x}, y) \sim p(\mathcal{D})$ **do**
4:     Randomly sample a parameter particle $\boldsymbol{\theta}_l \sim \boldsymbol{\Theta}$
5:     $\mathbf{x}_{\text{adv}} \leftarrow \mathbf{x}$
6:     **for** $t = 1 \rightarrow T$ **do**
7:         $\mathbf{x}_{\text{adv}} = \Pi_{\varepsilon_{\max}} \{\mathbf{x}_{\text{adv}} + \alpha \cdot \text{sign}\left(\nabla_{\mathbf{x}}\ell\left(f\left(\mathbf{x}_{\text{adv}}; \boldsymbol{\theta}_l\right), y\right)\right)\}$ {Generate Adversarial (Eq. (7))}
8:     **end for**
9:     **for** $i = 1 \rightarrow n$ **do**
10:         $\boldsymbol{\theta}_i \leftarrow \boldsymbol{\theta}_i - \epsilon_i \hat{\boldsymbol{\phi}}^*(\boldsymbol{\theta}_i, \boldsymbol{\theta}_l)$     with $\hat{\boldsymbol{\phi}}^*(\boldsymbol{\theta}_i, \boldsymbol{\theta}_l) = \sum_{j=1}^n \left[k(\boldsymbol{\theta}_j, \boldsymbol{\theta}_i)\nabla_{\boldsymbol{\theta}_j} L(\boldsymbol{\theta}_l) - \frac{\gamma}{n}\nabla_{\boldsymbol{\theta}_j} k(\boldsymbol{\theta}_j, \boldsymbol{\theta}_i)\right]$

11:         $\epsilon_i$ is the step size at the current iteration, $k(\boldsymbol{\theta}, \boldsymbol{\theta}')$ is a positive definite kernel that specifies the similarity between $\boldsymbol{\theta}$ and $\boldsymbol{\theta}'$, IG is the Information Gain (Eq. (8)), $\gamma, \lambda$ is the weight to control the *repulsive force* that enforces the diversity among parameter particles and IG objective respectively, $\ell$ is the cross-entropy loss function.
12:     **end for**
13: **end for**

---

### 3.5 A RELATION BETWEEN ADVERSARIAL AND OBSERVATIONAL TRAINING

A typical machine learning approach minimizes the empirical risk to learn. There are theoretical and empirical studies on the relation between the empirical risk and the true risk that measures the generalization ability of a learning algorithm. Generalization bounds such as Rademacher complexity or VC dimension for classical approaches or more recent studies for deep learning (see *e.g.* Neyshabur et al. (2017)) underpin the theoretical framework for machine learning. However, the relation between the risk when using samples from the observational distribution (*i.e.* the given dataset) versus when using their adversarial counterpart remains unexplored. It is important, because, while adversarial training has been commonly used, the impact of using such an approach on generalization with respect to the true data distribution is unknown. We particularly consider a Bayesian model with no specific assumption on the distribution of either the adversarial examples or the perturbations to provide a generic approach. The only major assumption we make for the following bound is that the distribution of the data and the corresponding adversarial are sufficiently close. That is a mild assumption when we consider the adversarial instances are obtained from small perturbations of the given training dataset.

To that end, we are interested in finding the bound of $|R_{\text{adv}} - R|$ where $R = \mathbb{E}_{\boldsymbol{\theta}}\left[\mathbb{E}_{(\mathbf{x},y)\sim\mathcal{D}}\left[\mathbb{E}_{y'\sim p(y|\mathbf{x},\boldsymbol{\theta})}\left[\mathbb{I}(y = y')\right]\right]\right]$ is the empirical risk and $R_{\text{adv}} = \mathbb{E}_{\boldsymbol{\theta}}\left[\mathbb{E}_{(\mathbf{x}_{\text{adv}},y)\sim\mathcal{D}_{\text{adv}}}\left[\mathbb{E}_{y'\sim p(y|\mathbf{x}_{\text{adv}},\boldsymbol{\theta})}\left[\mathbb{I}(y = y')\right]\right]\right]$ is the risk of the adversarial examples. Once we can obtain these, we can simply obtain the overall generalization and robustness bound. The following proposition summarizes our findings.

**Proposition 1.** *The risk of a classifier when trained on the observed training set denoted by $R$ versus when trained with adversarials denoted by $R_{adv}$ is bounded, i.e.*

$$|R_{adv} - R| \leq 1 - \mathbb{E}_{(\mathbf{x},y)\sim\mathcal{D}}\left[\exp\left(-\left(\mathbb{E}_{\boldsymbol{\theta}}[r_{\boldsymbol{\theta}}(\boldsymbol{x}, \boldsymbol{x}_{adv}, y)] + \lambda|\mathbb{E}_{\boldsymbol{\theta}}[IG(\boldsymbol{x}, y)] - \mathbb{E}_{\boldsymbol{\theta}}[IG(\boldsymbol{x}_{adv}, y)]|\right)\right)\right],$$

*where $r_{\boldsymbol{\theta}}(\boldsymbol{x}, \boldsymbol{x}_{adv}, y) = \sum_c^K p(y = c \mid \boldsymbol{x}, \boldsymbol{\theta})\log(p(y = c \mid \boldsymbol{x}_{adv}, \boldsymbol{\theta})), \lambda \geq 0$ and $\boldsymbol{x}_{adv}$ denotes the adversarial example obtained from $\boldsymbol{x}$.*

*Sketch of the Proof.* We simplify the difference between the risks by considering that the difference between individual mistakes is smaller than their product, *i.e.*

$$\mathbb{E}_{y_1 \sim p(y|\mathbf{x},\boldsymbol{\theta})} \left[ \mathbb{E}_{y_2 \sim p(y|\mathbf{x}_{\mathrm{adv}},\boldsymbol{\theta})} \left[ \mathbb{I}[y \neq y_1] - \mathbb{I}[y \neq y_2] \right] \right] \leq \mathbb{E}_{y' \sim p(y|\mathbf{x}_{\mathrm{adv}},\boldsymbol{\theta})} \left[ \mathbb{E}_{y' \sim p(y|\mathbf{x}_{\mathrm{adv}},\boldsymbol{\theta})} \left[ \mathbb{I}[y_1 \neq y_2] \right] \right]$$

$$\leq 1 - \sum_{c=1}^{K} p(y = c \mid \mathbf{x}, \boldsymbol{\theta}) p(y = c \mid \mathbf{x}_{\mathrm{adv}}, \boldsymbol{\theta}).$$

We then use Jensen's inequality when using the $\exp(\log(\cdot))$ to obtain the upper bound. The complete proof is provided in the Appendix B.

Then the difference between the empirical risk and the adversarial risk is minimized when the upper bound is minimized. The main objective is to:

1. Minimize cross entropy for the adversarial examples. This corresponds to matching the prediction from the adversarial data to that of the observations. Since $(\mathbf{x}, y)$ is given in the training, we simply minimize the entropy of the adversarial examples.

2. Minimize the difference between the information gained from the dataset and its adversarial counterparts. In addition to individual predictions, the information gained from each instance has to have a similar effect on both networks in terms of how it changes the parameters.

Notably, since we know $1 - \exp(-z) \leq z$, to avoid computational instabilities and gradient saturation, we consider minimizing the upper bound without the exponential function.

## 4 EXPERIMENTAL RESULTS

In this section, we verify the performance of our proposed method (IG-BNN) with other baselines in the literature on two popular and standard vision tasks: i) the low dimensional dataset CIFAR-10 (Krizhevsky et al.) This is a popular benchmark was used to evaluate the robustness of a DNN in previous works (Madry et al., 2018; Athalye et al., 2018). However, it is also known that adversarial training becomes increasingly hard for high dimensional data (Schmidt et al., 2018). Therefore, we evaluated our method on a high dimensional dataset—STL-10 (Coates et al., 2011) with 5,000 training images and 8,000 testing images with the dimension of $96 \times 96$ pixels. In all experiments, we utilized the same networks used in the state-of-the-art BNN method, Adv-BNN (Liu et al., 2019) to fairly compare the results. Specifically, we used the VGG-16 network architecture for CIFAR-10 and the smaller ModelA network for STL-10 used in Liu et al. (2019). The number of steps and the attack budgets used for training and testing is set to be the same for a fair comparison—see Appendix C Table 4. Because our proposed method evaluates the robustness of a Bayesian learning method based on Adversarial Training (Madry et al., 2018), the traditional Adversarial Training (Adv. Training) (Madry et al., 2018) and state-of-the-art Bayesian defense, Adv Bayesian Neural Network (Adv-BNN) (Liu et al., 2019) are good baselines for comparisons. In addition, we also compare our method with networks trained with no defenses and Bayesian Neural Networks trained for the tasks.

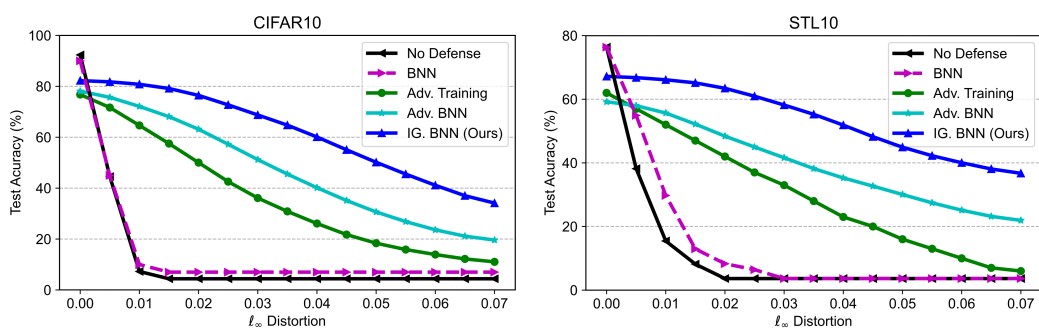

Figure 2: Accuracy under $\ell_\infty$-PGD attack on different datasets. CIFAR10 is trained on VGG-16 network, and STL10 is trained on ModelA similar to Adv-BNN

### 4.1 EVALUATE THE ROBUSTNESS UNDER WHITE-BOX $l_\infty$ ATTACK

**PGD attack**. In this experiment, we compare the robustness of our models under the strong white-box $l_\infty$-PGD attack. Following the recent work in (Liu et al., 2019), we set the maximum $l_\infty$ distortion to $\varepsilon_{\max} \in [0 : 0.07 : 0.005]$, adjust the PGD attacks for Bayesian methods as mentioned earlier–see Equation (3)—and report the accuracy on the test set (*robustness*). Overall, the results—shown in Figure 2 —illustrates the improved robustness of our method compared with Adv. BNN (Liu et al., 2019), and the significantly better results compared to Adv. Training (Madry et al., 2018). We also provide detailed results in Table 1 where: i) we show a marked increasing testing accuracy from approximately 10% to 20% compared with Adv-BNN; and a significantly higher accuracy compared with Adv. Training on the two tasks under increasing attack budgets. Although Adv-BNN helped improve robustness, we can see that the learning method is still below what could be achieved. On the other hand, IG-BNN achieved better results on both the testing data (benign) and adversarial examples (under increasing attack budgets).

Table 1: Comparing the robustness under different levels of PGD attacks (or attack budgets).

| Data | Defenses | 0 | 0.015 | 0.035 | 0.055 | 0.07 |
|---|---|---|---|---|---|---|
| | Adv. Training | 80.3 | 58.3 | 31.1 | 15.5 | 10.3 |
| CIFAR10 | Adv-BNN | 79.7 | 68.7 | 45.4 | 26.9 | 18.6 |
| | IG-BNN (Ours) | **82.2** | **79.7** | **65.6** | **46.1** | **32.6** |
| | Adv. Training | 63.2 | 46.7 | 27.4 | 12.8 | 7.0 |
| STL10 | Adv-BNN | 59.9 | 51.8 | 37.6 | 27.2 | 21.1 |
| | IG-BNN (Ours) | **67.0** | **65.6** | **57.0** | **45.3** | **36.7** |

**Proposed Adaptive PGD attack**. An adaptive attacker might find a better way to approach the real gradient under Bayesian context, hence, generate a stronger attack and degrade the robustness of a Bayesian Neural Network. Inspired from (Zimmermann, 2019), we proposed an Adaptive PGD attack where we aggregate the gradients across multiple random networks for each step. While this attack is slower because it needs inference from multiple particles or network instances for each PGD step, it is able to generate a stronger attack due to its better representative approximation to estimate the gradient. Specifically, we tailor a PGD attack for a Bayesian setting and use the expectation w.r.t to the parameter $\boldsymbol{\theta}$ to produce an adversarial example. That is, we can create an adversarial instance by taking multiple steps using the expected gradient across multiple models in each step towards maximizing the loss:

$$\mathbf{x}^{t+1} = \Pi_{\varepsilon_{\max}} \left\{ \mathbf{x}^t + \alpha \cdot \text{sign} \left( \mathbb{E}_{\boldsymbol{\theta}} \left[ \nabla_{\mathbf{x}} \ell \left( f \left( \mathbf{x}^t; \boldsymbol{\theta} \right), y_o \right) \right] \right) \right\} \tag{10}$$

The results for our proposed Adaptive PGD attack is shown in Table 2. As shown, the robustness of Adv-BNN has degraded significantly with the Adaptive PGD method although better than the Adv. Training method. On the other hand, our method, although losing some robustness compared with the previous PGD attack method, still achieves significantly higher robustness compared with other state-of-the-art defense methods in all the experimented vision tasks and across all attack budgets; thus, showing the effectiveness of our learning method.

Table 2: Comparing the robustness under different levels of Adaptive PGD attacks (or attack budgets).

| Data | Defenses | 0 | 0.015 | 0.035 | 0.055 | 0.07 |
|---|---|---|---|---|---|---|
| | Adv. Training | 80.3 | 58.3 | 31.1 | 15.5 | 10.3 |
| CIFAR10 | Adv-BNN | 79.7 | 64.2 | 37.7 | 16.3 | 8.1 |
| | IG-BNN (Ours) | **82.2** | **75.3** | **52.2** | **28.9** | **18.1** |
| | Adv. Training | 63.2 | 46.7 | 27.4 | 12.8 | 7.0 |
| STL10 | Adv-BNN | 59.9 | 47.9 | 31.4 | 16.7 | 9.1 |
| | IG-BNN (Ours) | **67.0** | **61.7** | **46.1** | **31.9** | **24.3** |

### 4.2 EVALUATE THE OBFUSCATED GRADIENT EFFECT

One possible failure mode of a defense methods discussed in the literature is the obfuscated gradient effect (Athalye et al., 2018) where seemingly high adversarial accuracy is only superficial and creates a false robustness. In this scenario, the network learns to obfuscate the gradients whilst showing

a seeming robustness by making it harder for the attack to find perturbations. However, an easy and effective way to verify this is to apply a black-box attack. The defense is considered to show obfuscated gradients if the black-box attack is more successful than the white-box attack. Following current practice, in this experiment, we deploy a black-box Square attack (Andriushchenko et al., 2020) on the our IG-BNN model. We can see in Table 3 that our IG-BNN is also highly robust against the black-box attack but the robustness of the black-box attack is higher than the white-box attack; this demonstrates our robustness is not simply the result of the obfuscated gradient effect.

Table 3: Blackbox attack to evaluate the obfuscated gradient effect

| Data | Defenses | 0 | 0.015 | 0.035 | 0.055 | 0.07 |
|---|---|---|---|---|---|---|
| CIFAR10 | IG-BNN (Ours) | 82.2 | 75.3 | 52.2 | 28.9 | 18.1 |
| | Black-box | - | 81.3 | 77.9 | 70.9 | 63.2 |
| STL10 | IG-BNN (Ours) | 67.0 | 61.7 | 46.1 | 31.9 | 24.3 |
| | Black-box | - | 67.0 | 64.4 | 62.7 | 58.4 |

### 4.3 Transfer attacks among parameter particles

To further evaluate the robustness and illustrate the intuition of the diverse parameter particles, we conduct experiments on the transferability of the adversarial examples among parameter particles and evaluate the robustness at class-wise levels (i.e. the robustness on each class). Specifically, we sample multiple different parameter particles for the experiment. For each parameter particle (*source particles*), we generate corresponding adversarial examples for that parameter particle. And then, using those adversarial examples generated from the source particles, attack and evaluate the robustness of other particles (*target particles*). We visualize the results as heatmaps with robustness as the measure (i.e. the ability to correctly identify the adversarial examples), and show the results in Figure 3 (comprehensive results are in the Appendix Section D). Each row in the matrix shows the robustness of target particles against the AEs generated from the source particles (with the attack budget $\epsilon = 0.015$).

We can observe that the adversarial examples are very effective on their source particles with 0% robustness. However, other particles are able to recognize those adversarial examples correctly (high robustness) due to our Bayesian learning method where we encourage the parameter particles to be diverse and bound the difference of empirical risk versus the adversarial risk in terms of the information gain.

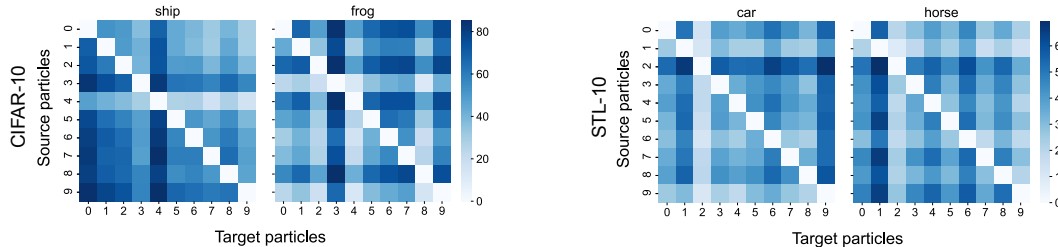

Figure 3: Diversity of parameter particles demonstrated using the transferability of adversarial examples among particles

## 5 Conclusion

In this study, we presented a novel method to learn a robust BNN against adversarial attacks. Although an adversarially trained BNN improved the robustness, using our proposed Adaptive PGD attack, tailored for BNN, can drastically reduce the robustness of adversarial-trained BNN. Our proposed IG-BNN learning method employing SVGD to encourage diverse parameter particles along with the formulated information gain under the Bayesian context to provably bound the difference of empirical risk versus adversarial risk. Through empirical experiments, we demonstrate that learning a Bayesian neural network using our method leads to better robustness compared with current state-of-the-art Bayesian defense methods. The learned model achieves robustness, even under strong Adaptive PGD attacks.

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

## A  DEFINITION OF INFORMATION GAIN

We first define our predictive distribution as:

$$p(y|\mathbf{x}, \mathcal{D}) = \int p(y|\mathbf{x}, \boldsymbol{\theta})p(\boldsymbol{\theta}|\mathcal{D})d\boldsymbol{\theta} \,.$$

By the definition of information gain, we have:

$$
\begin{aligned}
\mathbb{E}[\mathrm{IG}(\mathbf{x})] &= \sum_y p(y|\mathbf{x}, \mathcal{D})) \int \frac{p(y|\mathbf{x}, \boldsymbol{\theta})p(\mathcal{D}|\boldsymbol{\theta})p(\boldsymbol{\theta})}{p(\mathcal{D})p(y|\mathbf{x}, \mathcal{D})} \log\left(\frac{p(y|\mathbf{x}, \boldsymbol{\theta})}{p(y|\mathbf{x}, \mathcal{D})}\right) d\boldsymbol{\theta} \\
&= \frac{1}{p(\mathcal{D})} \sum \int p(y|\mathbf{x}, \boldsymbol{\theta})p(\mathcal{D}|\boldsymbol{\theta})p(\boldsymbol{\theta}) \log\left(\frac{p(y|\mathbf{x}, \boldsymbol{\theta})}{p(y|\mathbf{x}, \mathcal{D})}\right) d\boldsymbol{\theta} \\
&= \frac{1}{p(\mathcal{D})} \sum \int p(y|\mathbf{x}, \boldsymbol{\theta})p(\boldsymbol{\theta}|\mathcal{D}) \log\left(\frac{p(y|\mathbf{x}, \boldsymbol{\theta})}{p(y|\mathbf{x}, \mathcal{D})}\right) d\boldsymbol{\theta} \\
&= \frac{1}{p(\mathcal{D})} \sum \int p(y|\mathbf{x}, \boldsymbol{\theta})p(\boldsymbol{\theta}|\mathcal{D}) \left[\log(p(y|\mathbf{x}, \boldsymbol{\theta})) - \log(p(y|\mathbf{x}, \mathcal{D}))\right] d\boldsymbol{\theta} \\
&= \frac{1}{p(\mathcal{D})} \sum \left[\int p(y|\mathbf{x}, \boldsymbol{\theta})p(\boldsymbol{\theta}|\mathcal{D}) \log(p(y|\mathbf{x}, \boldsymbol{\theta}))d\boldsymbol{\theta} - \int p(y|\mathbf{x}, \boldsymbol{\theta})p(\boldsymbol{\theta}|\mathcal{D}) \log(p(y|\mathbf{x}, \mathcal{D}))d\boldsymbol{\theta}\right] \\
&= \frac{1}{p(\mathcal{D})} \int p(\boldsymbol{\theta}|\mathcal{D}) \sum p(y|\mathbf{x}, \boldsymbol{\theta}) \log(p(y|\mathbf{x}, \boldsymbol{\theta}))d\boldsymbol{\theta} - \sum \int p(y|\mathbf{x}, \boldsymbol{\theta})p(\boldsymbol{\theta}|\mathcal{D}) \log(p(y|\mathbf{x}, \mathcal{D}))d\boldsymbol{\theta} \\
&= \frac{1}{p(\mathcal{D})} \left(\mathbb{E}_{\boldsymbol{\theta}}[\mathbb{H}[y|\mathbf{x}, \mathcal{D}]] - \mathbb{H}[\mathbb{E}_{\boldsymbol{\theta}}[y|\mathbf{x}, \mathcal{D}]]\right) \\
&\propto \left(\mathbb{E}_{\boldsymbol{\theta}}[\mathbb{H}[y|\mathbf{x}, \mathcal{D}]] - \mathbb{H}[\mathbb{E}_{\boldsymbol{\theta}}[y|\mathbf{x}, \mathcal{D}]]\right)
\end{aligned}
$$

where for the last line we assume $p(\mathcal{D}) \approx p(\mathcal{D}_{\mathrm{adv}})$ as constant values. Since we are considering adversarial instances to be obtained from the observational one, this is a very mild assumption and is completely in line with current line of research.

## B  PROOF OF THE OBJECTIVE

We have

$$
\begin{aligned}
|R_{\mathrm{adv}} - R| &= \left|\mathbb{E}_{(\mathbf{x},y)\sim\mathcal{D}}\left[\mathbb{E}_{\boldsymbol{\theta}}\left[\sup \mathbb{E}_{y_1\sim p(y|\mathbf{x}_{\mathrm{adv}})}\left[\mathbb{I}(y_1 \neq y)\right] - \mathbb{E}_{y_2\sim p(y|\mathbf{x})}\left[\mathbb{I}(y_2 \neq y)\right]\right]\right]\right|, \\
&= \left|\mathbb{E}_{(\mathbf{x},y)\sim\mathcal{D}}\left[\mathbb{E}_{\boldsymbol{\theta}}\left[\sup \mathbb{E}_{y_1\sim p(y|\mathbf{x}_{\mathrm{adv}}),y_2\sim p(y|\mathbf{x})}\left[\mathbb{I}(y_1 \neq y) - \mathbb{I}(y_2 \neq y)\right]\right]\right]\right|, \\
&\leq \mathbb{E}_{(\mathbf{x},y)\sim\mathcal{D}}\left[\mathbb{E}_{\boldsymbol{\theta}}\left[\sup \mathbb{E}_{y_1\sim p(y|\mathbf{x}_{\mathrm{adv}}),y_2\sim p(y|\mathbf{x})}\left[|\mathbb{I}(y_1 \neq y) - \mathbb{I}(y_2 \neq y)|\right]\right]\right], \\
&\leq \mathbb{E}_{(\mathbf{x},y)\sim\mathcal{D}}\left[\mathbb{E}_{\boldsymbol{\theta}}\left[\sup \mathbb{E}_{y_1\sim p(y|\mathbf{x}_{\mathrm{adv}}),y_2\sim p(y|\mathbf{x})}\left[\mathbb{I}(y_1 \neq y_2)\right]\right]\right].
\end{aligned}
$$

where we can upper bound the expected misclassification to have:

$$\mathbb{E}_{(\mathbf{x},y)\sim\mathcal{D}}\left[\mathbb{E}_{\boldsymbol{\theta}}\left[\left[1 - \sum_{c=1}^K p(y = c \mid \mathbf{x}, \boldsymbol{\theta})p(y = c \mid \mathbf{x}_{\mathrm{adv}}, \boldsymbol{\theta})\right]\right]\right].$$

Subsequently, we use Jensen's inequality and the fact that $\mathbf{x} = \exp(\log(\mathbf{x}))$ to have:

$$\mathbb{E}_{(\mathbf{x},y)\sim\mathcal{D}}\left[\mathbb{E}_{\boldsymbol{\theta}}\left[\left[1 - \exp(\log(\underbrace{\sum_{c=1}^K p(y = c \mid \mathbf{x}, \boldsymbol{\theta})p(y = c \mid \mathbf{x}_{\mathrm{adv}}, \boldsymbol{\theta})}_{\geq\sum_c^K p(y=c|\mathbf{x},\boldsymbol{\theta})\log(p(y=c|\mathbf{x}_{\mathrm{adv}},\boldsymbol{\theta})}))\right]\right]\right],$$

and since $1 - \exp(z)$ is monotonically decreasing, we have

$$\mathbb{E}_{(\mathbf{x},y)\sim\mathcal{D}}\left[\mathbb{E}_{\boldsymbol{\theta}}\left[\left[1 - \exp(\log(\sum_{c=1}^{K} p(y = c \mid \mathbf{x}, \boldsymbol{\theta})p(y = c \mid \mathbf{x}_{\mathrm{adv}}, \boldsymbol{\theta})))\right]\right]\right]$$

$$\leq \mathbb{E}_{(\mathbf{x},y)\sim\mathcal{D}}\left[\mathbb{E}_{\boldsymbol{\theta}}\left[\left[1 - \exp\big(\sum_{c}^{K} p(y = c \mid \mathbf{x}, \boldsymbol{\theta})\log(p(y = c \mid \mathbf{x}_{\mathrm{adv}}, \boldsymbol{\theta}))\big)\right]\right]\right]$$

$$= 1 - \mathbb{E}_{(\mathbf{x},y)\sim\mathcal{D}}\left[\mathbb{E}_{\boldsymbol{\theta}}\left[\left[\exp\big(\sum_{c}^{K} p(y = c \mid \mathbf{x}, \boldsymbol{\theta})\log(p(y = c \mid \mathbf{x}_{\mathrm{adv}}, \boldsymbol{\theta}))\big)\right]\right]\right].$$

Therefore, we finally have the following bound:

$$|R_{\mathrm{adv}} - R| \leq 1 - \mathbb{E}_{(\mathbf{x},y)\sim\mathcal{D}}\left[\left[\exp\bigg(\mathbb{E}_{\boldsymbol{\theta}}\Big[\underbrace{\sum_{c}^{K} p(y = c \mid \mathbf{x}, \boldsymbol{\theta})\log(p(y = c \mid \mathbf{x}_{\mathrm{adv}}, \boldsymbol{\theta}))}_{r_{\boldsymbol{\theta}}(\mathbf{x},\mathbf{x}_{\mathrm{adv}},y)}\Big]\bigg)\right]\right].$$

This results show that the difference between the risks is bounded by the negative cross entropy of the predictions. While informative, this bound expresses the relation between the predictions only and not how the model perform on each set (*i.e.* given dataset versus its corresponding adversarial).

We know

$$\mathbb{E}_{\boldsymbol{\theta}}[r_{\boldsymbol{\theta}}(\mathbf{x}, \mathbf{x}_{\mathrm{adv}}, y)] \geq \mathbb{E}_{\boldsymbol{\theta}}[r_{\boldsymbol{\theta}}(\mathbf{x}, \mathbf{x}_{\mathrm{adv}}, y)] - |\mathbb{E}_{\boldsymbol{\theta}}[\mathrm{IG}(\mathbf{x})] - \mathbb{E}_{\boldsymbol{\theta}}[\mathrm{IG}(\mathbf{x}_{\mathrm{adv}})]| \tag{11}$$

or

$$\mathbb{E}_{\boldsymbol{\theta}}[r_{\boldsymbol{\theta}}(\mathbf{x}, \mathbf{x}_{\mathrm{adv}}, y)] = -\mathbb{E}_{\boldsymbol{\theta}}[\mathrm{KL}(p(y = c \mid \mathbf{x}, \boldsymbol{\theta})\|p(y = c \mid \mathbf{x}_{\mathrm{adv}}, \boldsymbol{\theta}))] + \mathbb{E}_{\boldsymbol{\theta}}[\mathbb{H}[y \mid \mathbf{x}, \mathcal{D}]$$
$$\geq -\mathbb{E}_{\boldsymbol{\theta}}[\mathrm{KL}(p(y = c \mid \mathbf{x}, \boldsymbol{\theta})\|p(y = c \mid \mathbf{x}_{\mathrm{adv}}, \boldsymbol{\theta}))] - |\mathbb{E}_{\boldsymbol{\theta}}[\mathrm{IG}(\mathbf{x})] - \mathbb{E}_{\boldsymbol{\theta}}[\mathrm{IG}(\mathbf{x}_{\mathrm{adv}})]| \tag{12}$$

where this is true because we can subtract positive values, namely $\mathbb{E}_{\boldsymbol{\theta}}[\mathbb{H}[y \mid \mathbf{x}, \mathcal{D}]$ and $|\mathbb{E}_{\boldsymbol{\theta}}[\mathrm{IG}(\mathbf{x})] - \mathbb{E}_{\boldsymbol{\theta}}[\mathrm{IG}(\mathbf{x}_{\mathrm{adv}})]|$. Hence we have for $\lambda \geq 0$,

$$|R_{\mathrm{adv}} - R| \leq 1 - \mathbb{E}_{(\mathbf{x},y)\sim\mathcal{D}}\left[\exp\bigg(\mathbb{E}_{\boldsymbol{\theta}}\Big[\sum_{c}^{K} p(y = c \mid \mathbf{x}, \boldsymbol{\theta})\log(p(y = c \mid \mathbf{x}_{\mathrm{adv}}, \boldsymbol{\theta}))\Big] - \lambda|\mathbb{E}_{\boldsymbol{\theta}}[\mathrm{IG}(\mathbf{x})] - \mathbb{E}_{\boldsymbol{\theta}}[\mathrm{IG}(\mathbf{x}_{\mathrm{adv}})]|\bigg)\right],$$

or

$$|R_{\mathrm{adv}} - R| \leq 1 - \mathbb{E}_{(\mathbf{x},y)\sim\mathcal{D}}\left[\exp\bigg(-\big(\mathbb{E}_{\boldsymbol{\theta}}[\mathrm{KL}(p(y = c \mid \mathbf{x}, \boldsymbol{\theta})\|p(y = c \mid \mathbf{x}_{\mathrm{adv}}, \boldsymbol{\theta}))] + \lambda|\mathbb{E}_{\boldsymbol{\theta}}[\mathrm{IG}(\mathbf{x})] - \mathbb{E}_{\boldsymbol{\theta}}[\mathrm{IG}(\mathbf{x}_{\mathrm{adv}})]|\big)\bigg)\right].$$

Then the difference between the empirical risk and the adversarial risk is minimized when the upper bound is minimized. The main objective is to:

1. Maximize $\mathbb{E}_{\boldsymbol{\theta}}[\sum_{c}^{K} p(y = c \mid \mathbf{x}, \boldsymbol{\theta})\log(p(y = c \mid \mathbf{x}_{\mathrm{adv}}, \boldsymbol{\theta}))]$ or minimize $\mathbb{E}_{\boldsymbol{\theta}}[\mathrm{KL}(p(y = c \mid \mathbf{x}, \boldsymbol{\theta})\|p(y = c \mid \mathbf{x}_{\mathrm{adv}}, \boldsymbol{\theta}))$: this corresponds to matching the prediction from the adversarial data to that of the observations. Since $(\mathbf{x}, y)$ is given in the training, for minimizing this KL-divergence we simply minimize the entropy of the adversarial examples instead;

2. Minimize $\mathbb{E}_{\boldsymbol{\theta}}[\mathrm{IG}(\mathbf{x})] - \mathbb{E}_{\boldsymbol{\theta}}[\mathrm{IG}(\mathbf{x}_{\mathrm{adv}})]$: In addition to individual prediction, the information gained from each instance has to have similar effect on both networks in terms of how it changes the parameters.

Notably, since we know $1 - \exp(-z) \leq z$, to avoid computational instabilities and gradient saturation, we consider minimizing the upper bound without the exponential function in our implementation.

# C    HYPER-PARAMETERS

Table 4: Hyper-parameters setting in our experiments

| Name | Value | Notes |
|---|---|---|
| $T'$ | 20 | #PGD iterations in attack at test time |
| $T$ | 10 | #PGD iterations in adversarial training |
| $\varepsilon_{\max}$ | 8/255 | Max $l_\infty$-norm in adversarial training |
| $\alpha$ | 2/255 | Step size for each PGD iteration |
| $\gamma$ | 0.01 | Weight to control the repulsive force |
| $\lambda$ | 5 | Weight to control IG objective |
| $n$ | 10 | #Forward passes when doing ensemble inference |

# D    DETAILS OF TRANSFER ATTACKS OF ADVERSARIAL EXAMPLES AMONG PARAMETER PARTICLES

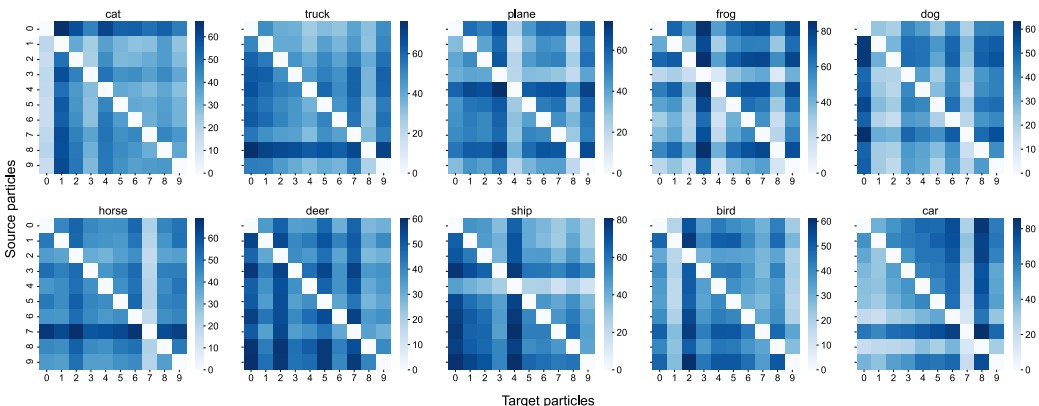

Figure 4: Transferability of adversarial examples among different particles on CIFAR-10

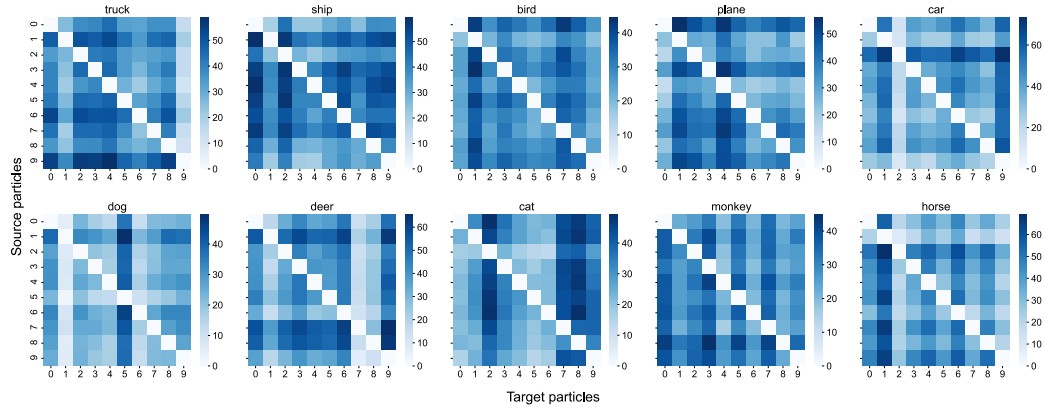

Figure 5: Transferability of adversarial examples among different particles on STL-10

