# OpenReview forum: "Bayesian Learning with Information Gain Provably Bounds Risk for a Robust Adversarial Defense"
_ICLR.cc/2022/Conference — ICLR 2022 Submitted_

### Official Review · Reviewer_CwGg · 2021-11-01

**Correctness:** 3
**Technical Novelty And Significance:** 2
**Empirical Novelty And Significance:** 3
**Recommendation:** 5
**Confidence:** 5

**Main Review:**

In general, the paper is clearly written by addressing an important problem, but I still have several concerns.

1.	It is interesting that the authors propose to use the Bayesian formulation for adversarial training. Nevertheless, the authors are expected to make a thorough analysis since the benign examples and the adversarial examples are not independent. The authors are expected to clarify the reasonability of Eq. 5.
2.	To some extent, the paper is highly dependent on the information gain. However, it is nontrivial to estimate the information gain especially for the adversarial scenarios. A neural network is always overconfident for the adversarial examples. The authors are expected to clarify to how to estimate the information gain reliably.
3.	The paper is inherently a regularized version for BNN with information gain. The authors are expected to clarify the novelties and technical contributions.
4.	In the experimental section, the performance is significantly better than the alternative methods, e.g., Adv-BNN, especially for the adaptive attacks. The authors are expected to make thorough analysis on why or for what technical aspects it can achieve the performance improvement.
5.	The authors are expected to make more comprehensive analysis, e.g., more experiments on ImageNet, more defense/attacks methods.



**Summary Of The Paper:**

In this paper, the authors propose to learn a multi-modal posterior of the Bayesian neural network to defense the adversarial attacks, which can prevent the collapse and encourage the diversity accordingly. The authors further investigate the information gain for adversarial Bayesian learning, which is used to guide the training a BNN with the information gain bound. Experimental results demonstrate tis superior performance for adversarial attacks.

**Summary Of The Review:**

The paper is easy to follow, but the authors are expected to clarify their technical contributions and make more comprehensive experiments.

---

> ### Author Response · Authors · 2021-11-23
> **Response to Reviewer CwGg**
>
> > __C5__. The authors are expected to make more comprehensive analysis, e.g., more experiments on ImageNet, more defense/attacks methods.
>
> __Response__: Thank you for your feedback. Let us clarify why our results are more extensive than published studies and why no study has used ImageNet.
>
> 1. Recently published work in [1] on the robustness of Bayesian Neural Networks was evaluated on simple datasets such as MNIST (input shape: 1x28x28), FashionMNIST (1x28x28)
> 2. Work in [2] was evaluated on similar simple datasets such as MNIST (input shape: 1x28x28), FashionMNIST (1x28x28) with the additional CIFAR-10 (3x32x32)
> 3. We evaluated on two datasets, similar to Adv-BNN, which are CIFAR-10 (3x32x32) and a higher dimensional dataset STL-10 (3x96x96).
>
> Further, in contrast to previous studies,
>
> 1. We have also implemented the black-box attacks in Section 4.2 to verify the robustness of our method in that scenario, to show that our robustness is not merely due to the obfuscated gradient effect (Athalye et al., 2018).
> 2. We also conducted transferable attacks among particles in Section 4.3 to show the intuition and robustness of our method at class-wise levels.
>
> The reason that BNN studies often limit evaluations to a few datasets (often with smaller image sizes) is because it takes considerable computational resources to train a BNN compared to a CNN. For example, Izmailov et al. (2021) utilized a cluster of 512 TPUs to run Hamiltonian Monte Carlo (HMC) for CIFAR-10 dataset.
>
> [1] Carbone, Ginevra, et al. "Robustness of Bayesian neural networks to gradient-based attacks." NeurIPS (2020).
>
> [2] Wicker, Matthew, et al. "Bayesian inference with certifiable adversarial robustness." International Conference on Artificial Intelligence and Statistics (AISTATS)., 2021.

---

> ### Author Response · Authors · 2021-11-23
> **Response to Reviewer CwGg**
>
> > __C1__. It is interesting that the authors propose to use the Bayesian formulation for adversarial training. Nevertheless, the authors are expected to make a thorough analysis since the benign examples and the adversarial examples are not independent. The authors are expected to clarify the reasonability of Eq. 5.
>
> __Response__: Thank you for your valuable feedback. We agree that the benign and adversarial examples are not independent. The assumption in the Eq. 5 is that adversarial examples are independent of each other, not independent from their benign counterpart. *We will clarify this assumption in the revised version.*
>
> > __C2__. To some extent, the paper is highly dependent on the information gain. However, it is nontrivial to estimate the information gain especially for the adversarial scenarios. A neural network is always overconfident for the adversarial examples. The authors are expected to clarify to how to estimate the information gain reliably.
>
> __Response__: Thank you for your valuable comment. We will clarify these points further in the revised version, till then we can provide the explanation below.
>
> We estimate information gain by sampling particles from the posterior distribution (Equation (5)) using SVGD inference method and compute the entropies henceforth.
>
> Different from traditional adversarial training methods where the network is trained only on adversarial examples (hence overconfident for adversarial examples), in this work, we propose measuring the Information Gain for both adversarial and benign examples. Then, the information gained from benign and adversarial examples is encouraged to be the same, which deters the model from being biased towards learning from the adversarial instances alone and enables the receptive fields to be active for similar and prominent patterns (in both benign and adversarial). These two objectives combined, namely, Information Gain and Bayesian setting, reduce the overconfidence of the neural network for the adversarial examples and improve the generalization performance.
>
> > __C3__. The paper is inherently a regularized version for BNN with information gain. The authors are expected to clarify the novelties and technical contributions.
>
> __Response__: Our novelties and contributions:
>
> - We have obtained a new state-of-the-art result for Adversarial Bayesian Learning under a rigorous evaluation regime.
>
> - Our approach, for the first time, formalizes adversarial learning from an information-theoretic perspective in which the information gain (IG) criterion is utilized with the intuition that the information content learned from the benign and adversarial instances are identical. We show this reduces the common issue of overconfidence in adversarial instances.
>
> - We provide an upper bound on the risk of learning from benign and adversarial instances. We showed that this bound has implications from Bayesian (i.e. the distribution of the parameters) and information-theoretic (i.e. the patterns the model recognizes for benign and adversarials) perspectives.
>
> - We employed SVGD for inference in BNNs and showed that it provides a significantly better inference approach for robustness in deep learning. This further highlights the need for better inference methods that capture the multimodal nature of the posterior and relies less on a single parameter choice.
>
> - With our extensive experiments, we show our approach achieves significantly better robustness compared to its counterparts; essentially, our efforts culminate in providing a new state-of-the-art result.
>
> > __C4__. In the experimental section, the performance is significantly better than the alternative methods, e.g., Adv-BNN, especially for the adaptive attacks. The authors are expected to make a thorough analysis on why or for what technical aspects it can achieve the performance improvement.
>
> __Response__: There are three key factors contributing to our performance compared with Adv-BNN:
>
> - We propose a better way to learn the Bayesian posterior distribution using SVGD with a repulsive force (Section 3.1)
> - We enforce an Information Gain formulation to avoid overconfidence of neural networks on adversarial examples.
> - We employ  a better and more efficient way to generate adversarial examples in the context of Bayesian adversarial learning, while the one used in Adv-BNN is shown to attain unrepresentative gradient directions as shown in Section 3.2.

---

### Official Review · Reviewer_UwZA · 2021-11-02

**Correctness:** 2
**Technical Novelty And Significance:** 2
**Empirical Novelty And Significance:** 2
**Recommendation:** 3
**Confidence:** 4

**Main Review:**

Strengths:
* The method appears to offer significant improvements over prior adversarial BNN defense.
* The authors implement an appropriate adaptive attack to account for the randomness of Bayesian Neural Networks. It might be helpful to refer to the attack with the widely-used terminology of Expectation-Over-Transformation (EOT) attack, as in Athalye et. al (2018). However, the authors place much more prominence in the text on a non-adaptive PGD attack, which might cause misperceptions (see Weaknesses).

Weaknesses:
* While the authors perform an appropriate adaptive EOT attack, the plots and main results present the outcomes of a non-adaptive PGD attack. It is well known that deterministic attacks are not representative of the performance of defenses that have stochastic elements, and the authors indeed see a significant drop in accuracy. Since a defense is only as good as its performance against an optimal attack, the authors should also include a curve showing robustness for their method against adaptive attacks in Figure 2 to give fair comparison versus adversarial training.
* The theoretical analysis appears to be quite trivial. Unless I am mistaken, adding the term $|E_\theta [ IG(x) ] - E_\theta [ IG(x_\text{adv}) ] |$ in (11) only makes the bound looser and thus less useful. The theoretical presentation does not enhance the arguments of the paper.
* The work lacks novelty, and its novel elements are not properly evaluated. The main contributions are the inclusion of SVGD and the Information Gain term. The theoretical importance of the Information Gain term is not very convincing. The authors do not perform an ablative study using just SVGD and no IG term vs. Gaussian variational inference plus IG term to narrow down which elements are important, or if both are necessary.
* The adaptive attack is not completely explored. The authors do not mention how many EOT replicates are used. 20 PGD steps is not enough for the EOT defense performance to reach its full strength in some cases where there is a high degree of stochasticity, and more steps should be used to ensure convergence.

Other Comments:
* How are the particles $\theta_i^0$ initialized during the testing phase? I would assume they are initialized from a set of fixed parameters identified during the training phase, is that correct? More details on this point would enhance my understanding of the work.
* Source code would help to further evaluate this defense.

**Summary Of The Paper:**

This paper explores adversarial training for Bayesian Neural Networks. In contrast to prior work which used Gaussian variational posteriors to estimate BNN parameters during adversarial training, this work uses Stein Variational Gradient Descent (SVGD) on a loss potential that includes an adversarial term similar to standard adversarial training and a term encouraging similarity between the Information Gain (IG) on natural and adversarial images. They evaluate PGD and adaptive adversarial attacks and find significant performance improvement over standard adversarial training and adversarial BNN. A theoretical justification is presented.

**Summary Of The Review:**

I recommend to not accept this paper. The theoretical analysis of the Information Gain term appears vacuous and distracting, although I could be missing something. The contributions (SVGD and Information Gain for BNN posterior sampling) are not significantly novel, and their relative importance is not properly evaluated. The experimental results appear strong, but it's difficult to judge the quality of the defense experiments without being able to inspect the to source code (in particular, the implementation of the EOT attack).

---

> ### Author Response · Authors · 2021-11-23
> **Response to Reviewer UwZA**
>
> > __C1__. While the authors perform an appropriate adaptive EOT attack, the plots and main results present the outcomes of a non-adaptive PGD attack. It is well known that deterministic attacks are not representative of the performance of defenses that have stochastic elements, and the authors indeed see a significant drop in accuracy. Since a defense is only as good as its performance against an optimal attack, the authors should also include a curve showing robustness for their method against adaptive attacks in Figure 2 to give fair comparison versus adversarial training.
>
> __Response__: We thank you for your valuable feedback.  Our results are in Table 2 but we will draw the curve for the EoT attack.
>
>
> > __C2__. The theoretical analysis appears to be quite trivial. Unless I am mistaken, adding the term in (11) only makes the bound looser and thus less useful. The theoretical presentation does not enhance the arguments of the paper.
>
> __Response__: Let us clarify:
>
> 1. The analysis in the paper shows how the risk for training using benign instances and the adversarials are related. We are the __*first to provide such a connection and discussion on its implications*__. Our bound shows that the generalization risk is bound by the expected cross entropy of the models (note that would be w.r.t the posterior distribution).
> 2. The IG provides an information theoretic constraint that intuitively ensures the model learns similar patterns in either benign or adversarial settings.
> 3. Furthermore, IG ensures that if there is an additional instance added to the model, then the update for both benign and adversarial is the same.
>
> > __C3__. The work lacks novelty, and its novel elements are not properly evaluated. The main contributions are the inclusion of SVGD and the Information Gain term. The theoretical importance of the Information Gain term is not very convincing. The authors do not perform an ablative study using just SVGD and no IG term vs. Gaussian variational inference plus IG term to narrow down which elements are important, or if both are necessary.
>
> __Response__: Thank you for your feedback. Our contribution is a new method to learn a BNN robust to adversarial examples. To do this, we propose the adversarial example generation, training as well as the inference method for a new Bayesian Learning framework to build a robust model with the incorporation of an Information Gain formulation.
>
> Our extensive results clearly demonstrate a __*new state-of-the-art result in the field*__; significantly better than published results and the results are generated from a careful evaluation. Hopefully our clarifications above provides the confidence that our adversarial training method, evaluation method as well as other settings are correct. We have also provided the codebases for independent validation of the results and the correctness of the method we employed. We agree, re-formulating the method by removing aspects is interesting but it does not change the outcome we have achieved and demonstrated.
>
> > __C4__. The adaptive attack is not completely explored. The authors do not mention how many EOT replicates are used. 20 PGD steps is not enough for the EOT defense performance to reach its full strength in some cases where there is a high degree of stochasticity, and more steps should be used to ensure convergence.
>
> __Response__: Let us clarify further here.
>
> 1. We use EOT over 10 sampled particles.
> 2. The reason we used 20 PGD steps is that we want a fair comparison with the related work where the authors also evaluate the robustness against attacks with 20 PGD steps.
> 3. Importantly, we only trained on weak PGD adversarials (PGD steps = 10), and evaluated on stronger PGD attacks (PGD steps = 20).
> Now we have evaluated our method against PGD steps=100 to explore the suggestion by the reviewer and show the result below (TableA). The results show that, even when we significantly increased the PGD steps, the robustness did not change much. This new result further validates our proposed method and demonstrates its effectiveness.
>
> **TableA. Evaluation of the robustness under different EoT PGD steps**
>
> | Data   | Defenses       | 0    | 0.015 | 0.035 | 0.055 | 0.07 |
> |--------|----------------|------|-------|-------|-------|------|
> | STL-10 | Ours (PGD-20)  | 67.0 | 61.7  | 46.1  | 31.9  | 24.3 |
> |        | Ours (PGD-100) | 67.0 | 61.8  | 45.7  | 30.6  | 22.9 |
> |        |                |      |       |       |       |      |
>
>
> > __C5__. How are the particles initialized during the testing phase? I would assume they are initialized from a set of fixed parameters identified during the training phase, is that correct? More details on this point would enhance my understanding of the work. Source code would help to further evaluate this defense.
>
> __Response__:  Yes, the particles are initialized based on the fixed parameters identified from the training phase. We will release the source code for clarity.

---

### Official Review · Reviewer_qjBg · 2021-11-03

**Correctness:** 2
**Technical Novelty And Significance:** 2
**Empirical Novelty And Significance:** 2
**Recommendation:** 3
**Confidence:** 4

**Main Review:**

**Strengths of the paper**
- The information gain is useful to improve the robustness of Bayesian Neural Networks.
- The proposed method outperforms the baselines.

**Weaknesses of the paper**
- Although the proposed method is intuitive, its framework is not solidly and rigorously developed.
- Information gain seems identical to the mutual information $I(y,\theta \mid x, D)$ proposed and investigated in Bayesian Active Learning [1,2]. The authors need to discuss their proposed information gain and the mutual information proposed in [1,2].

[1] Houlsby, Neil, et al. "Bayesian active learning for classification and preference learning." arXiv preprint arXiv:1112.5745 (2011).

[2]  Yarin Gal, Riashat Islam, and Zoubin Ghahramani. Deep Bayesian active learning with image data. In Proceedings of the 34th International Conference on Machine Learning-Volume 70, pages 1183–1192. JMLR. org, 2017.

**Summary Of The Paper:**

This paper proposes leveraging "information gain" with Bayesian Neural Networks to improve the robustness of Bayesian Neural Networks. Specifically, it maintains a set of particle models and uses SVGD to update this set of particle models. Furthermore, it uses the PGD-based attack to a specific particle model to craft adversarial examples and enforces the information gain of adversarial examples to be close to that of the corresponding benign examples.

**Summary Of The Review:**

The posterior $p(\theta \mid D_{adv})$ in Eq. (5) is not mathematically rigorous because it does not contain $D$ on the left side, but $D$ is involved on the right side.

I cannot see the definition of the information gain and it seems to be identical to the mutual information proposed in [1,2]. Moreover, the first term on the right side of  Eq. (8) is $H(y \mid x, D)$ because it is independent with $\theta$. The second term should be $E_\theta[H(y \mid x, \theta)]$ according to your derivation in Appendix.

SVGD is not proposed to minimize the loss function directly. Although the update of the particle models in Algorithm 1 is fine, it requires more effort to define the distribution for which we only know its unnormalized version. Furthermore, Eq. (9) needs to have the absolute value for the difference between $IG(x)$ and $IG(x_{adv})$. In addition, the notion here is inconsistent with the lack of $y$.

---

> ### Author Response · Authors · 2021-11-23
> **Response to Reviewer qjBg**
>
> > __C1__. Information gain seems identical to the mutual information proposed and investigated in Bayesian Active Learning [1,2]. The authors need to discuss their proposed information gain and the mutual information proposed in [1,2].
>
> __Response__: Thank you for raising this point; we will clarify further here and also update the paper.
>
> The derivation of the information gain is that in [1,2]. Our contribution is the use of IG concepts in the context of adversarial learning (our proposed formulation in Bayesian Learning) because of the clear intuition it provides: the information content that the model recognizes from the benign and adversarial samples should be the same.
>
> We provided the derivation of IG in the Appendix for completeness, but we will clarify this further in the revised version.
>
> > __C2__. Typos in Eq (5), (8) and (9)
>
> __Response__: Thank you for your thorough review and for pointing out these typos in the paper. We have fixed all of them and will update in the revision.
>
> > __C3__. SVGD is not proposed to minimize the loss function directly. Although the update of the particle models in Algorithm 1 is fine, it requires more effort to define the distribution for which we only know its unnormalized version.
>
> __Response__: We are sorry for the lack of clarity. Since the inference approach of SVGD and its proof has been rigorously investigated in Wang & Liu (2019), Liu & Wang (2016); we rely on these prior works in Algorithm 1. We will clarify this further in the revised version.

---

> > ### Comment · Reviewer_qjBg · 2021-11-29
> > **Feedback to reviewers**
> >
> > Thanks for your responses to me. I believe that this paper needs a major improvement. Therefore, I keep my score unchanged.

---

### Official Review · Reviewer_RZv2 · 2021-11-03

**Correctness:** 2
**Technical Novelty And Significance:** 1
**Empirical Novelty And Significance:** 2
**Recommendation:** 3
**Confidence:** 4

**Details Of Ethics Concerns:**

None for this paper.

**Main Review:**

Unfortunately, there are many claims made in this paper which are either overly strong, incorrect, or fail to appreciate prior work.

Firstly, the authors correctly note that Bayesian neural networks are empirically and theoretically more robust to attacks. I think the authors should certainly cite [1] here as a motivating example.

The authors highlight the work of Adv-BNN as state of the art for Bayesian neural networks against adversarial attacks, but this has been greatly surpassed in [2] (more on this below). Moreover, their "proposed adaptive PGD attack" is not their own proposal and has been used in the Bayesian robustness literature for at least 3 years. The reason this proposed attack has been used is that the one given in Adv-BNN and repeated in equations (3) and (7) of this paper is both theoretically incorrect and empirically suboptimal. As pointed out by the authors, [3] clearly states the incorrectness of the attack used in Adv-BNN and shows it is suboptimal in practice for Bayesian neural networks (further corroborated by Tables 1 and 2 of this paper). This error is the Bayesian counter-part to the expectation over transformations (EoT) attack formalized in [4] which follows up on a work the authors cite, in particular [5]. Further, in works on BNNs this "proposal" has been adopted and is widely used, see [1,2,6], so I think it is incorrect to deem this as a "proposed" attack in the evaluation/contribution section of the paper and should be moved to the related works.

Further on this, despite the clear incorrectness of equation (3) and its corresponding formulation for this paper in equation (7) according to both deterministic principles [4,5] and Bayesian principles [2,3] the authors still use this attack during training and partially during evaluation. Thus, the methodology, as presented, is suboptimal/incorrect. In fact, the results section of this paper corroborate this claim (Table 2 versus Table 1)

Outside of this incorrectness, the authors propose a regularization which is simply a specific form of Adversarial Logit Pairing [7]. This paper originally appeared in NeurIPS 2018, but was later retracted after it was found that this kind of robustness is very easily attacked and provides no substantial gains see [8]. I think if the authors would like to stick to this methodology then they ought to properly evaluate their method with a Bayesian adaptation of the method in [8]. I do not doubt the theoretical perspective given by the authors, only that it is practically effective/worthwhile. Moreover, I think the authors ought to be very careful in their general use of ERM approaches to likelihoods. This is perhaps acceptable for variational inference methods which replace marginalization with maximization, but in general, the method of [2] is preferable as it works for both maximization and marginalization and is thus on better grounds from a Bayesian point of view. Further on this point, I think it would benefit the paper if the authors stated their algorithm in a more general form as simply proposing a modification of a single inference method reduces the apparent contribution.

Finally, I do have a further question about the methods evaluation. Is this method compared with AdvBNN applied to SVGD? Or is AdvBNN using Bayes by Backprop (as in its original formulation)? If so, the authors should ensure that the same inference method is used between their method and that of the network they compare against as inference method is known to have a considerably effect on the robustness of Bayesian posteriors [1,2,6]. In fact, the empirical performance noted in this paper might be solely the effect of a more faithful inference method and may have no correlation with the use of their method. Unless, the "BNN" label in Figure 2 is also SVGD. Still, the authors ought to compare with [2] prior to making the claim that they have state of the art robustness.

[1] - https://arxiv.org/abs/2002.04359 (Appeared in NeurIPS last year)
[2] - https://arxiv.org/abs/2102.05289 (Appeared in AISTATS last year)
[3] - https://arxiv.org/abs/1907.00895 (Correction to Adv-BNN in 2019)
[4] - https://arxiv.org/pdf/1707.07397.pdf (Appeared in ICML 2018)
[5] - https://arxiv.org/pdf/1802.00420.pdf (Already cited in paper but here for completeness)
[6] - https://arxiv.org/abs/2012.12640 (Appeared in AABI 2021)
[7] - https://arxiv.org/abs/1803.06373 (Appeared, later retracted NeurIPS 2018)
[8] - https://arxiv.org/abs/1807.10272



**Summary Of The Paper:**

In this paper, the authors propose a conjunction of Bayesian learning and regularization via information gain as a means of adversarial defense. In particular, they start from the perspective that Bayesian learning of neural network parameters ought to be more robust to adversarial examples, and then state that by jointly minimizing the risk and regularizing adversarial and clean information gain that the resulting Bayesian posterior will be even more robust.

While this -- robustness and adversarial training of Bayesian neural networks -- is a worthwhile and important direction, the authors make several crucial mistakes and literature oversights which inhibit the potential impact of this work.

**Summary Of The Review:**

The paper is in a worthwhile direction, but ultimately has too many oversights and errors to be accepted as is. The attack methodology relied upon is known to be incorrect and suboptimal. The regularization introduced is a particular flavor of a defense which is known to be suboptimal and easily attacked. The claim of state-of-the-art performance is outdated for BNNs.

---

> ### Author Response · Authors · 2021-11-23
> **Response to Reviewer RZv2**
>
> > __C8__. I do not doubt the theoretical perspective given by the authors, only that it is practically effective/worthwhile. Moreover, I think the authors ought to be very careful in their general use of ERM approaches to likelihoods. This is perhaps acceptable for variational inference methods which replace marginalization with maximization, but in general, the method of [2] is preferable as it works for both maximization and marginalization and is thus on better grounds from a Bayesian point of view. Further on this point, I think it would benefit the paper if the authors stated their algorithm in a more general form as simply proposing a modification of a single inference method reduces the apparent contribution.
>
> __Response__: Thank you for your valuable suggestion. While we agree that our approach is more generic and is not bound to a particular inference method, we highlight that in fact there are practical concerns that make SVGD more attractive for this problem. Our experiments have shown that given the high-dimensional parameter space of the deep neural networks, SVGD performs the best in finding reasonably diverse instances that lead to better robustness (please see Figure 1, Figure 3 and Table 2).
>
> > __C9__. Finally, I do have a further question about the methods evaluation. Is this method compared with AdvBNN applied to SVGD? Or is AdvBNN using Bayes by Backprop (as in its original formulation)? If so, the authors should ensure that the same inference method is used between their method and that of the network they compare against as inference method is known to have a considerably effect on the robustness of Bayesian posteriors [1,2,6]. In fact, the empirical performance noted in this paper might be solely the effect of a more faithful inference method and may have no correlation with the use of their method. Unless, the "BNN" label in Figure 2 is also SVGD.
>
> __Response__: We agree and as with our discussion in Section 3.1 and 4.1 we expect the use of SVGD to improve robustness. However, this has not been shown before. Hence we contribute towards confirming the expectation, also highlighted by the reviewer.
>
> In summary, our work highlights that for  BNNs to be successfully applied to adversarial learning, we need an inference method that faithfully captures the posterior distribution that SVGD provides a good solution for. We will follow the reviewer's suggestion to conduct the ablative study to show the effectiveness of our method.

---

> ### Author Response · Authors · 2021-11-23
> **Response to Reviewer RZv2**
>
> > __C4__. The reason this proposed attack has been used is that the one given in Adv-BNN and repeated in equations (3) and (7) of this paper is both theoretically incorrect and empirically suboptimal. As pointed out by the authors, [3] clearly states the incorrectness of the attack used in Adv-BNN and shows it is suboptimal in practice for Bayesian neural networks (further corroborated by Tables 1 and 2 of this paper). This error is the Bayesian counter-part to the expectation over transformations (EoT) attack formalized in [4] which follows up on a work the authors cite, in particular [5].
>
> __Response__: Thank you for your valuable comments. We agree with the reviewer that the adversarial example generated using equation (3) is suboptimal (we highlighted this in our paper in Section 3.2). That is why we used Equation (7).
>
> The adversarial example generation we describe in Equation (7) is not the same as that in (3); although at first, the equations do look very similar. So, we disagree that the approach in Equation (7) is the same as Equation (3) used in Adv-BNN. We will clarify in details below.
>
> In the adversarial example generation in (7), a randomly sampled $\theta$ from the Bayesian posterior distribution is used for all of the PGD attack steps for *each* input instance x (i.e. the whole PGD process will be deployed on a given instance of a neural network or a deterministic neural network).
>
> This is different from Equation (3) where a $\theta$ is randomly sampled from the Bayesian posterior distribution for each PGD attack step. Hence,  the resulting gradient direction is likely to be unrepresentative.
>
> The adversarial example generation described in Equation (7) has some similarities with the approach in the recommended work in [2]. In Section 4 of the recommended work in [2], the authors also find the robust likelihood for each single deterministic Neural Network (NN) (sampled from the BNN) at a time, this is essentially the approach we used in Equation (7).
>
> > __C5__. Further, in works on BNNs this "proposal" has been adopted and is widely used, see [1,2,6], so I think it is incorrect to deem this as a "proposed" attack in the evaluation/contribution section of the paper and should be moved to the related works.
>
> __Response__: We are sorry for the poor use of language here. We included some background in the proposal for completeness. However, we agree with the reviewer that moving it to the related work may improve clarity and will do so in the next version.
>
> > __C6__. Further on this, despite the clear incorrectness of equation (3) and its corresponding formulation for this paper in equation (7) according to both deterministic principles [4,5] and Bayesian principles [2,3] the authors still use this attack during training and partially during evaluation. Thus, the methodology, as presented, is suboptimal/incorrect. In fact, the results section of this paper corroborate this claim (Table 2 versus Table 1)
>
> __Response__: Please see our responses in C4 where we clarify: i) the subtle but important differences in Equation (3) used in Adv-BNN and ours in Equation (7) during training, and ii) the similarity of Equation (7) to the approach used in [2]. So we respectfully disagree that the training method is suboptimal/incorrect.
>
> Further, this is quantitatively demonstrated in Table 2 in our paper. We show our approach outperforms its counterparts, with a robustness gap of up to 15%. This result supports our claims in our response to C4 that this approach using Equation (7) improves robustness.
>
> > __C7__. Outside of this incorrectness, the authors propose a regularization which is simply a specific form of Adversarial Logit Pairing [7]. This paper originally appeared in NeurIPS 2018, but was later retracted after it was found that this kind of robustness is very easily attacked and provides no substantial gains see [8]. I think if the authors would like to stick to this methodology then they ought to properly evaluate their method with a Bayesian adaptation of the method in [8].
>
> __Response__:  Thank you for your valuable suggestions. The Adversarial Logit Pairing [7] enforces the logits of adversarial and benign inputs to be close in Euclidean distance, which is quite different from our approach.
>
> As opposed to the heuristic that was considered in reference [7] we provide an information-theoretic objective that simplifies into a regularization. We politely disagree with the reviewer that a simple heuristic is the same as such a rigorous undertaking for addressing the problem.

---

> ### Author Response · Authors · 2021-11-23
> **Response to Reviewer RZv2**
>
> > __C1__. Firstly, the authors correctly note that Bayesian neural networks are empirically and theoretically more robust to attacks. I think the authors should certainly cite [1] here as a motivating example.
>
> __Response__: Thank you for your valuable suggestion, we will cite [1] as a motivating example.
>
> > __C2__. The authors highlight the work of Adv-BNN as state-of-the-art for Bayesian neural networks against adversarial attacks, but this has been greatly surpassed in [2] (more on this below).
>
> __Response__: Thanks for your recommendation. Indeed, [2] is an interesting work and we will surely cite this paper in the related work. However, we can highlight the following regarding [2].
>
> 1. In [2], for the same dataset (CIFAR-10) used in our paper, the certified robustness is only valid for the attack budget range of $\epsilon=1/255=0.004$ and up to 1.57 in the Supplemental Material. This attack budget is quite small (i.e. a weak attack). In contrast, our defense is more focused on strong attacks with the various evaluations against attack budgets up to 0.07; this value is 17 times stronger than what was evaluated in [2] (please see Table 2 in our paper).
> 2. The evaluation in [2] is for the same attack budget $\epsilon$ used in training, while ours is evaluated against not only the training $\epsilon$ budget but also on stronger attack budgets. This makes our defense more reflective of a real-world deployment scenario where the attacker can craft an adversarial with a larger attack budget.
> 3. Further, even though [2] did not evaluate robustness under the same setting we employed, [2] followed the Interval Bound Propagation (IBP) method of [refA], and the evaluation in [refA] showed that training a certified CIFAR-10 network using IBP with an increasing value of $\epsilon$ will lead to a significant degradation in the clean accuracy. For example, in [refA] with the same training budget of $\epsilon=8/255$, the clean accuracy of the CIFAR-10 model dropped from 83.34% to 47.14%. Thus, training a certified network on strong attack budgets comes with a significant trade-off in benign accuracy.
> 4. Unfortunately, the link to the Github repository provided in the paper [linkB] no longer exists and we are not able to generate quantitative results for comparison. Even with the current results reported in the paper, albeit for a different network and different adversarial training method, we believe that we cannot make a meaningful comparison; for one, our benign accuracy on the CIFAR-10 task is 82% while the result reported in [2] is around 60%.
>
> Thus, we feel the published version of Adv-BNN, albeit with the known short-comings but with published code and models, still provides the best method for a quantitative comparison.
>
> [refA] Gowal, Sven, et al. "On the effectiveness of interval bound propagation for training verifiably robust models." arXiv preprint arXiv:1810.12715 (2018).
>
> [linkB] https://github.com/matthewwicker/CertifiableBayesianInference
>
>
> > __C3__. Moreover, their "proposed adaptive PGD attack" is not their own proposal and has been used in the Bayesian robustness literature for at least 3 years.
>
> __Response__: We are sorry for the unclear statement here. Notably, we did not take credit for the adaptive attack as mentioned in Section 4.1,  and cited the work in [3] that proposed the correct method to apply an  expectation over transformations (EoT) attack to Bayesian neural networks.

---

### Decision · Program_Chairs · 2022-01-20

**Decision:**

Reject

**Comment:**

In this paper, the authors leverage information gain in conjunction with Bayesian Neural Networks in order to to improve the robustness of Bayesian Neural Networks. However, as pointed out by reviwers, there are several mistakes in theier derivations and evaluations. Moreover, the authors failed to crrectly refer to the exisiting work proposing similar methods.